# Supervised Exercise in Water: Is It a Viable Alternative in Overweight/Obese People with or without Type 2 Diabetes? A Pilot Study

**DOI:** 10.3390/nu14234963

**Published:** 2022-11-23

**Authors:** Roberto Pippi, Matteo Vandoni, Matteo Tortorella, Vittorio Bini, Carmine Giuseppe Fanelli

**Affiliations:** 1Healthy Lifestyle Institute, C.U.R.I.A.Mo. (Centro Universitario Ricerca Interdipartimentale Attività Motoria), Department of Medicine and Surgery, University of Perugia, Via G. Bambagioni, 19, 06126 Perugia, Italy; 2Laboratory of Adapted Motor Activity (LAMA), Department of Public Health, Experimental Medicine and Forensic Science, University of Pavia, 27100 Pavia, Italy; 3Guy’s and St Thomas’ NHS Foundation Trust, Great Maze Pond, London SE1 9RT, UK; 4Department of Medicine and Surgery, University of Perugia, Via Gambuli, 1, 06132 Perugia, Italy

**Keywords:** exercise, overweight/obesity, type 2 diabetes, water-based exercise

## Abstract

The study of the effects of a water-based exercise program in overweight/obese people with or without type 2 diabetes is a topic of relatively recent interest. This type of exercise presents some advantages in reducing the risk of injury or trauma, and it can be a valuable therapeutic card to play for sedentary or physically inactive patients who have chronic metabolic diseases. This work aims to make a contribution showing the effects of a water-based exercise intervention, supervised by graduates in sports sciences, in a group of overweight/obese people with or without type 2 diabetes. In total, 93 adults (age 60.59 ± 10.44 years), including 72 women (age 60.19 ± 10.97 years) and 21 men (age 61.95 ± 8.48 years), were recruited to follow a water-based exercise program (2 sessions/week, for 12 weeks) at the C.U.R.I.A.Mo. Healthy Lifestyle Institute of Perugia University. Results showed an improvement in body mass index (−0.90 ± 1.56, *p* = 0.001), waist circumference (−4.32 ± 6.03, *p* < 0.001), and systolic (−7.78 ± 13.37, *p* = 0.001) and diastolic (−6.30 ± 10.91, *p* = 0.001) blood pressure. The supervised water-based intervention was useful in managing patients with metabolic diseases who often present with other health impairments, such as musculoskeletal problems or cardiovascular or rheumatic disease that could contraindicate gym-based exercise.

## 1. Introduction

Type 2 diabetes (DM2) is an ever-increasing, noncommunicable disease usually associated with overweight, obesity, and a sedentary lifestyle. The beneficial effects of physical activity and exercise in people with overweight/obesity with or without DM2 are well documented by solid evidence [1,2,3,4]. Guidelines from the World Health Organization (WHO) [5], the American College of Sports Medicine (ACSM) [6], and the American Diabetes Association (ADA) [7], as well as Italian national standards of care [8], state that adult people should be engaged in at least 150–300 min per week of moderate aerobic physical activity or at least 75–150 min of vigorous aerobic physical activity, or an equivalent combination, in addition to muscle-strengthening activities on two or more days per week [5]) to obtain substantial health benefits. Despite this, many people are sedentary or physically inactive (people who do not engage in at least 150 min per week of MVPA are defined as inactive by WHO), with increased health risks [9]. Physical activity level and sedentary behaviors affect people’s health status [10], especially people with obesity and DM2 [11], who frequently are also affected by one or more osteoarticular or micro- or macroangiopathic complications.

In water-based exercise, the effect of body weight is limited due to the partial weight support the water provides. This in turn reduces weight-bearing stress on skeletal joints [12]. As a result, people with overweight/obesity can more easily engage in water-based exercise rather than ground-based exercise, which can cause more physical discomfort due to the effect of gravity [13]. By reducing the risk of injury or trauma, aquatic exercise can be a valuable therapeutic card to play in place of total weight-bearing exercises or aerobic activity, such as walking on land [14], which patients are less likely to enjoy and, in some cases, are associated with greater risk of injury or long-term disability. According to Bonifazi (2005), water-based exercise is a “type of activity that includes performing exercises to improve muscle tone and joint mobility” [15].

Although some authors have highlighted how physical activities practiced in water are useful in managing patients with health impairments such as musculoskeletal damage, cardiovascular diseases, and rheumatic and metabolic diseases [16,17,18,19,20,21], the effect of a water-based exercise intervention on health has not been comprehensively studied in people with overweight or obesity with or without DM2. In particular, the efficacy in weight reduction and metabolic improvements is not clear/well documented.

For these reasons, this work aims to show the effects of a water-based supervised exercise intervention on weight management and metabolic panel in overweight/obese people with or without DM2.

## 2. Materials and Methods

### 2.1. Participants

Among 1464 patients enrolled between January 2010 and February 2014 in the C.U.R.I.A.Mo. trial a total sample (Figure 1) of 93 adults (mean age of 60.59 ± 10.44 years), including 72 women (age 60.19 ± 10.97 years) and 21 men (age 61.95 ± 8.48 years), was recruited to follow a multidisciplinary intervention at the C.U.R.I.A.Mo. Healthy Lifestyle Institute of Perugia University. The intervention included educational classes with endocrinologists and nutritionists as well as a water-based exercise program (2 sessions/week, for 12 weeks). Particularly at the beginning, the exercise program entailed water-based drills which were aimed to improve both patients’ relationship with water and their swimming technique [22].

Inclusion criteria were: age between 37 and 78 years; BMI ≥ 25 kg/m^2^; the presence of musculoskeletal disorders or other health conditions that could contraindicate gym exercise or Nordic walking activities. The exclusion criteria were: the presence of clinical conditions that could seriously reduce life expectancy or ability to participate in the study; no written informed consent to study participation. According to these criteria, this study involved 90 adults (mean age of 60.86 ± 9.50 years), including 69 women (age 60.52 ± 9.82 years) and 21 men (age 61.95 ± 8.48 years). Furthermore, a selection was performed to include only participants (*n* = 38 overweight/obese people, of which *n* = 20 with diabetes and 18 without diabetes) with baseline (T0) and follow-up (T1) data on every measured variable. Finally, we studied a small group of participants (*n* = 23 overweight/obese people, of which *n* = 13 with diabetes and 10 without diabetes) who presented all data for anthropometric and clinical variables at T0, T1, and T2.

### 2.2. Study Design

This is anuncontrolled pilot study involving a subsample of subjects who participated in the C.U.R.I.A.Mo. (Centro Universitario di Ricerca Interdipartimentale Attività Motoria) trial [22]. The C.U.R.I.A.Mo. trial has been registered in the Australian New Zealand Clinical Trials Registry (a Primary Registry in the WHO registry network) with the number ACTRN12611000255987. Anthropometric and clinical as well as other haematobiochemical variables were assessed at the beginning (T0), and after three months (T1) of the water-based exercise program using a quasiexperimental study design. For a small group of participants (*n* = 23, those patients who have adhered to all the follow-up visits foreseen by the C.U.R.I.A.Mo. program), data after one year (T2) from the end of the intensive period of activity are also available.

### 2.3. The C.U.R.I.A.Mo. Clinical Model

All the participants have been assessed through the C.U.R.I.A.Mo. evaluation model composed of some clinical steps: (1) a clinical examination, conducted by an endocrinologist, to exclude the existence of any clinical conditions that could contraindicate the exercise and to prescribe blood sample tests according to national standards of care [8]; (2) a brief motivational interview, conducted by a psychologist, to assess participants psychological status; (3) a nutritional assessment conducted to assess the nutritional habits of the participants; (4) a water-based exercise intervention consisting of three-month repeated cycles of water exercises, as specified below.

### 2.4. The Exercise Intervention

The exercise program was a water-based exercise intervention in a 25 m swimming pool lane. The protocol consisted of a 12-week water-based exercise program of two sessions per week lasting fifty (50) minutes each.

Each session was composed of an initial warm-up of twenty (20) minutes, in which each patient performed light water-based movements and some breathing exercises. The warm-up procedures were adapted to each participant’s skill level and involved swimming unaided or with aids (noodles, floating belts). Each participant was asked to start slowly and increase exercise intensity subjectively to a moderate intensity during which, ideally, they could still talk to each other.

The second part of the session was resistance-oriented and based on upper and lower limb exercises lasting fifteen (15) minutes. During this section, upper and lower limbs were exercised through water-resistance movements. Those participants that showed improvement through the weeks had extra resistance added by introducing floating dumbbells for the upper limbs and noodles for the lower limbs.

After the resistance section, a cool-down of ten (10) minutes and some light stretching for five (5) minutes were provided. During the cool-down, each participant was required to execute some light and continuous water exercise at a lower intensity than the warm-up. Stretching was executed in the water and involved mainly upper and lower limbs.

After the first follow-up medical visit, we offered the patients the opportunity to repeat the three-month cycle of water-based exercise.

### 2.5. Clinical Assessments

Before (T0) starting, at the end of the three-month exercise program (T1) and after one year (T2), all the participants underwent a clinical assessment of anthropometric and clinical variables, as follows:

#### 2.5.1. Anthropometric Variables

Anthropometric measurements of weight, body mass index (BMI), and waist circumference (WC) were collected. Weight was assessed using a Tanita body composition analyzer BC-420MA (Tokyo, Japan), and BMI was calculated as weight (kg)/height (m)^2^. According to the World Health Organization and the International Diabetes Federation (IDF) tips [23], WC was measured in the horizontal plane midway between the lowest ribs and the iliac crest, using the ergonomic circumference measuring tape Seca^®^ 201 model.

#### 2.5.2. Clinical Variables

Systolic (SBP) and diastolic (DBP) blood pressure and blood sample variables, such as fasting blood glucose, glycated hemoglobin (HbA1c), total cholesterol, high-density lipoprotein (HDL) cholesterol, low-density lipoprotein (LDL) cholesterol, triglycerides, and uric acid were also collected.

Systolic blood pressure (SBP) and diastolic blood pressure (DBP) values were measured three times during the first medical visit with one minute between consecutive measurements 5 min of sitting at rest using an upper arm blood pressure monitor (UM-101 mercury-free sphygmomanometer, A&D Medical, Naples, Italy) on the right upper arm using a size-appropriate cuff.

Biochemical blood test variables were collected from tests performed by patients in routine clinical practice.

### 2.6. Statistical Analysis

Descriptive analyses in terms of means, standard deviations, or percentages were computed for each investigated variable at baseline (T0). Participants’ data were then split into two subgroups according to the pathological condition diagnosed by the endocrinologist (overweight/obesity with or without DM2), and an unpaired t-test was conducted to compare all variables at baseline (please see Appendix A). To evaluate the effects of the water-based exercise program, parameters at baseline and after three months of exercise intervention were compared through a t-test for paired samples (Table 1). Delta (Δ) changes (T1-T0) are presented as means ± standard deviations. Moreover, to evaluate the effects of water-based exercise programs over a longer period (1 year), parameters at baseline, after 3 (T1), and after 12 months (T2) were compared through a linear mixed model repeated measures analysis of variance in a small group of participants (*n* = 23) who presented all data for anthropometric and clinical variables collected both before and after the intervention. Means ± standard errors for all data and delta changes (using a method of multiple testing correction with Bonferroni adjustment) were presented (Table 2). *p*-value  <  0.05 was indicated as statistically significant. Finally, to study the delta changes trend of declining, we calculated the B coefficient with SE for all the variables using linear regression with time as the independent variable. All analyses were performed with IBM SPSS^®^ version 25.0 (IBM Corp. Released 2013. IBM SPSS Statistics for Windows, Version 22.0. Armonk, NY, USA: IBM Corp.).

## 3. Results

### 3.1. Baseline Results

Anthropometric and clinical variables values did not show statistically significant differences between overweight/obesity with or without DM2 at baseline (please see Appendix A), except for fasting blood glucose (*p* = 0.003), HbA1c (*p* = 0.004), and LDL cholesterol (*p* = 0.005).

### 3.2. Follow-Up Results

At the first follow up (T1), statistically significant improvements in weight (*p* = 0.013) and BMI (*p* = 0.001) and reductions in WC (*p* < 0.001), SBP (*p* = 0.001), and DBP (*p* = 0.001) were observed in the entire sample (Table 1). Clinical effects, although not statistically significant, were observed in the differential values of blood sample variables such as fasting blood glucose (−1.49 mg/dL), HbA1c (−0.28%), total cholesterol (−8.00 mg/dL), HDL (+1 mg/dL), LDL cholesterol (−8.55 mg/dL), triglycerides (−8.62 mg/dL), and uric acid (−0.10 mg/dL). In the overweight/obese with diabetes, data showed statistically significant variations in weight (*p* = 0.037) and BMI (*p* = 0.008) and reductions in WC (*p* < 0.001), SBP (*p* = 0.028), and DBP (*p* = 0.002), while in patients without diabetes, we observed reductions in SBP (*p* = 0.016) and fasting blood glucose (*p* = 0.039).

Data analyzed from all patients with an available one-year (T2) follow-up evaluation (*n* = 23, 1 male and 22 female, mean age 58.78 ± 9.17 years) confirmed a statistically significant change (Table 2) in the following outcomes: weight (*p* = 0.023), BMI (*p* = 0.020), WC (*p* < 0.001), SBP (*p* = 0.013), and DBP (*p* = 0.019). Furthermore, the time factor effect was observed in total cholesterol (*p* = 0.013) and LDL (*p* = 0.024).

## 4. Discussion

This study aimed to show the effects of a water-based exercise program of 24 biweekly sessions on some anthropometric (i.e., weight, BMI, and WC) and clinical (i.e., SBP and DBP, lipids, and glycemic variables) outcomes in a group of overweight/obese adults with and without DM2. Preliminary results showed an improvement in BMI, WC, SBP, and DBP in all the participants that exercised for 12 weeks.

Previously, contrasting results were obtained in similar water-based exercise programs. A Tanaka’s review article [24] reported that few studied have addressed the effects of swimming exercise on bodyweight and them stated no changes in weight [25,26]. Recent studies conducted by Rezaeipour et al. [27] and Cugusi et al. [2] observed a significant reduction in weight and BMI, as we observed in this study. Conversely, Rezaeipour, M. (2020) [28] reported that pool workouts are ineffective in terms of weight loss and changes in body composition, as also stated by Gubiani et al. [29]. In an epidemiological study, other authors [30] observed that regular physical activity (including swimming exercise) is associated with an attenuation of weight regain over a middle–long period (10 years). Interestingly, our study results showed that weight reduction was observed not only after a short period of a water-based exercise program but also after 1 year (respectively, −3.12 kg and −6.64 kg).

Previous studies showed contrasting metabolic effects of water-based exercise training. Some authors evaluated anthropometric and cardiometabolic parameters in women [31,32] and adult men with obesity and DM2 [2]. While the first observed no significant improvements in body weight and body composition, Cugusi et al. showed a significant improvement in the cardiovascular system and metabolic profile (body weight, WC, BMI, and blood pressure improvements). Some authors [33,34,35] have studied the impact of regular exercise on lipid and lipoprotein plasma levels, especially HDL, as the main tool for reducing the risk of cardiovascular diseases. Other authors [36] have specifically studied the effects of water exercise on blood lipid outcomes. Rezaeipour et al. [33] found that there was not a significant alteration of blood lipid parameters, and we did not observe a statistically significant improvement in total cholesterol, HDL, LDL cholesterol, and triglycerides (*p* > 0.005), after the intensive phase of the exercise program (12 weeks). These results could probably be due to the short period of observation and nutritional habits. In fact, at T2 (Table 2) data seems to show encouraging results regarding lipid control.

Water-based exercise effects on blood pressure were assessed in previous studies. A recent meta-analysis [37] reported that water-based exercise had positive effects on BP values. Particularly, SBP decrease was estimated equal to −8.4 mmHg, while DBP decrease was estimated equal to −3.3 mmHg. After three months of water-based exercise in our sample, we found a lower improvement in SBP (−7.78 mmHg), but a higher improvement of DBP (−6.30 mmHg). Although Delevatti et al. [38] concluded that “aerobic training in an aquatic environment provides effects similar to aerobic training in a dry-land environment in patients with type 2 diabetes”, the result of the present study seems to be promising compared to previous studies conducted in small samples of people with overweight/obesity who exercised for three months in a gym-based program [39].

Certainly, this study has some limitations. First, we did not include a control group that could have better clarified the extent of the results; the original C.U.R.I.A.Mo. intervention model did not foresee it. Second, we enrolled only a small group of participants. This was due mainly to technical and methodological reasons. It was possible to provide access to the pool for only a small number of patients for each exercise group. Both for the limited space available and for the need to guarantee personalized supervision of the activity, we decided to set up groups no larger than five participants each. Additionally, a water exercise-based program encountered various critical issues in the adherence of patients, who have reported difficulties related to the time necessary to carry out activities in the water (i.e., long preparation times due for example to the fact of having to dry the hair, etc.). Furthermore, many key measures (i.e., waist circumference, blood pressure) were derived using manual methods subject to investigator bias, even if measurements at T0 and T1 were always conducted by the same operator to minimize operator variability and all operators were specifically trained in accordance with standard measurement procedures. Additionally, in this study, we did not show data referring to nutritional habits and pharmacological treatment. In future studies, an analysis of eating habits should be carried out together with insights into the drugs taken, given that these components are relevant in this type of patient with metabolic diseases. Our results showed that in a relatively short period of educational training it is possible to obtain positive effects which have a significant impact on people’s health. Reductions in anthropometric (such as BMI and WC) and clinical (such as blood pressure and cholesterol) variables values are linked to obesity and cardiovascular risks [40]. These aspects are particularly important for people with health conditions for whom weight-bearing exercise is contraindicated or for the elderly [41,42].

## 5. Conclusions

The supervised water-based intervention was effective in improving weight, BMI, waist circumference, and blood pressure. Although it showed lower results on short-term glucometabolic control, this type of exercise practiced in water was useful in managing patients with health problems—i.e., musculoskeletal problems, cardiovascular disease, rheumatic and metabolic diseases—that could contraindicate gym-based exercises. Longer periods of intervention and observation, which may involve larger numbers of participants, are desirable for future research.

## Figures and Tables

**Figure 1 nutrients-14-04963-f001:**
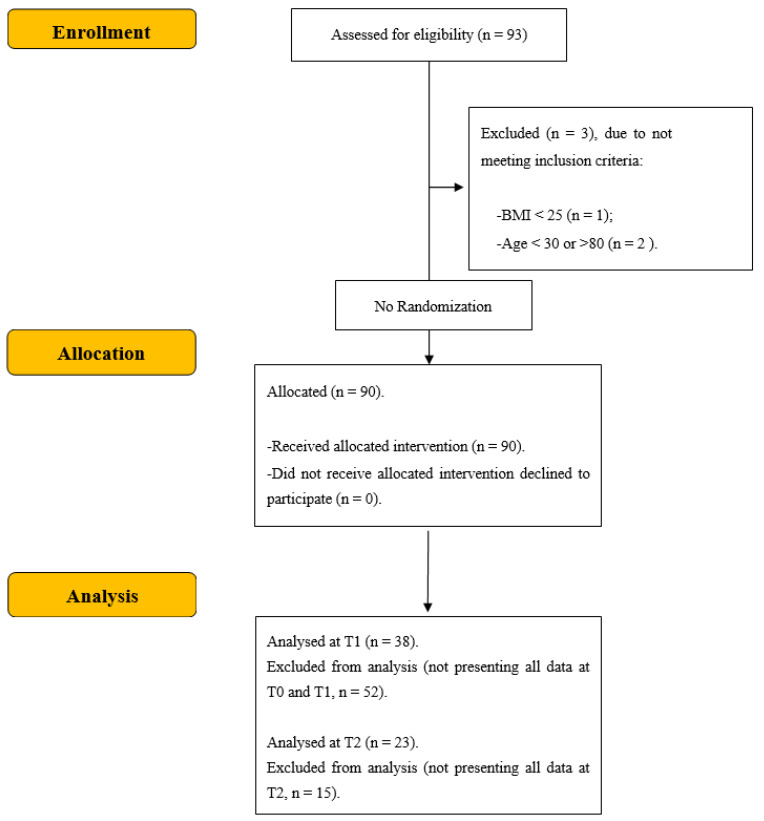
CONSORT flow diagram (adapted from CONSORT 2010 flow diagram).

**Table 1 nutrients-14-04963-t001:** Anthropometric, blood pressure, and blood chemistry parameters values at baseline (T0) in the entire sample (*n* = 38), overweight/obese with diabetes (*n* = 20), and overweight/obese without diabetes (*n* = 18) subjects. Results are presented as mean ± standard deviation. Statistical significance was set for *p* values ≤ 0.05.

Variables	All			Overweight/Obese with Diabetes			Overweight/Obese without Diabetes		
	T0	Δ Test-T (T1 vs. T0)	*p*	T0	Δ Test-T (T1 vs. T0)	*p*	T0	Δ Test-T (T1 vs. T0)	*p*
Weight (kg)	65.96 ± 47.62	−2.61 ± 7.79	0.013	72.59 ± 46.99	−4.16 ± 10.45	0.037	58.87 ± 48.08	−1.00 ± 2.71	0.057
BMI (kg/m^2^)	36.77 ± 6.21	−0.90 ± 1.56	0.001	37.44 ± 7.25	1.24 ± 1.8	0.008	36.03 ± 4.89	−0.54 ± 1.22	0.075
WC (cm)	115.82 ± 12.89	−4.32 ± 6.03	<0.001	118.85 ± 15.63	5.55 ± 5.69	<0.001	112.44 ± 8.11	−2.94 ± 6.25	0.062
SBP (mmhg)	136.08 ± 11.85	−7.78 ± 13.37	0.001	136.75 ± 13.11	−8.0 ± 15.08	0.028	135.29 ± 10.53	−7.53 ± 11.48	0.016
DBP (mmhg)	81.89 ± 7.67	−6.30 ± 10.91	0.001	82.5 ± 7.86	9.15 ± 11.18	0.002	81.18 ± 7.61	−2.94 ± 9.85	0.236
Fasting blood glucose (mg/dL)	108.95 ± 30.37	−1.49 ± 20.91	0.668	122.1 ± 36.09	1.10 ± 27.48	0.860	94.33 ± 11.02	−4.53 ± 8.29	0.039
HbA1c (%)	6.36 ± 1.22	−0.28 ± 0.85	0.074	6.91 ± 1.46	−0.33 ± 1.08	0.206	5.78 ± 0.48	−0.22 ± 0.41	0.083
Total cholesterol (mg/dL)	211 ± 51.41	−8.00 ± 35.43	0.184	197.70 ± 57.34	−5.26 ± 36.89	0.542	225.78 ± 40.48	−11.06 ± 34.59	0.206
HDL (mg/dL)	49.45 ± 9.79	1.00 ± 9.93	0.550	49.25 ± 10.65	0.47 ± 11.76	0.863	49.67 ± 9.04	1.59 ± 7.71	0.408
LDL cholesterol (mg/dL)	132.41 ± 42.30	−8.55 ± 27.67	0.072	114.72 ± 43.62	−2.34 ± 26.68	0.707	152.07 ± 31.48	−15.49 ± 6.76	0.036
Triglycerides (mg/dL)	152.68 ± 82.58	−8.62 ± 53.19	0.331	169.65 ± 92.95	−17.05 ± 63.27	0.243	133.83 ± 66.88	1.29 ± 37.66	0.889
Uric acid (mg/dL)	5.64 ± 1.36	−0.10 ± 0.78	0.510	5.71 ± 1.44	0.02 ± 0.77	0.942	5.58 ± 1.33	−0.22 ± 0.80	0.349

Legend: BMI = body mass index; WC = waist circumference; SBP = systolic blood pressure; DBP = diastolic blood pressure; HbA1c = glycosylated haemoglobin; Hdl = high-density lipoprotein; LDL = low-density lipoprotein.

**Table 2 nutrients-14-04963-t002:** Anthropometric, blood pressure, and blood chemistry parameters values at baseline (T0) and after 3 months (T1) and 12 months (T2) post-intervention. Results are presented as mean ± standard errors (SE) in *n* = 23 subjects. Statistical significance was set for *p* values ≤ 0.05.

Variables	T0	T1 (3 Months)	T2 (12 Months)	*p*			Delta Changes					
					B	SE						
							T1 vs. T0	*p*	T2 vs. T0	*p*	T2 vs. T1	*p*
Weight (kg)	58.70 ± 7.14	55.57 ± 7.00	52.06 ± 7.13	0.023	−1.517	2.382	−3.12 ± 1.21	0.038	−6.64 ± 3.05	0.103	−3.52 ± 2.83	0.659
BMI (kg/m^2^)	37.27 ± 1.18	35.98 ± 1.19	36.1 ± 1.17	0.020	−0.585	0.827	−1.28 ± 0.34	0.003	−1.17 ± 0.58	0.169	0.11 ± 0.51	1.000
WC (cm)	117.22 ± 2.28	111.52 ± 2.35	110.09 ± 2.32	<0.001	−3.727	2.563	−5.70 ± 0.97	<0.001	−7.13 ± 1.06	<0.001	−1.43 ± 0.56	0.050
SBP (mmhg)	135.95 ± 2.20	129.14 ± 2.82	126.48 ± 2.37	0.013	−4.435	1.604	−6.81 ± 3.32	0.160	−9.48 ± 3.24	0.025	−2.67 ± 2.86	1.000
DBP (mmhg)	82.62 ± 1.68	77 ± 1.61	76.71 ± 1.24	0.019	−2.952	1.095	−5.62 ± 2.66	0.141	−5.90 ± 1.99	0.023	−0.29 ± 2.05	1.000
Fasting blood glucose (mg/dL)	117.83 ± 6.85	114.78 ± 6.47	120.57 ± 9.11	0.500	−0.955	4.877	−3.04 ± 4.18	1.000	2.74 ± 5.88	1.000	5.78 ± 4.39	0.604
HbA1c (%)	6.63 ± 0.32	6.24 ± 0.22	6.30 ± 0.23	0.099	−0.167	0.186	−0.40 ± 0.21	0.234	−0.33 ± 0.21	0.395	0.06 ± 0.06	0.894
Total cholesterol (mg/dL)	222.75 ± 11.46	208.92 ± 8.76	201.17 ± 8.79	0.013	−10.457	7.086	−13.83 ± 7.30	0.212	−21.58 ± 7.42	0.024	−7.75 ± 4.24	0.242
HDL (mg/dL)	48.29 ± 1.97	49.12 ± 2.45	49.17 ± 2.52	0.097	0.587	1.690	0.83 ± 2.23	1.000	0.88 ± 2.43	1.000	0.04 ± 2.05	1.000
LDL (mg/dL)	140.34 ± 9.98	129.38 ± 7.73	123.97 ± 7.39	0.024	−7.848	6.153	−10.96 ± 5.87	0.224	−16.38 ± 6.39	0.052	−5.42 ± 3.28	0.336
Triglycerides (mg/dL)	171.42 ± 19.35	152.04 ± 18.17	139.96 ± 11.40	0.066	−16.522	12.212	−19.38 ± 11.96	0.357	−31.46 ± 14.96	0.140	−12.08 ± 12.47	1.000
Uric acid (mg/dL)	5.49 ± 0.33	5.37 ± 0.30	5.55 ± 0.24	0.612	0.031	0.201	−0.12 ± 0.21	1.000	0.06 ± 0.16	1.000	0.18 ± 0.16	0.866

Legend: BMI = body mass index; WC = waist circumference; SBP = systolic blood pressure; DBP = diastolic blood pressure; HbA1c = glycosylated haemoglobin; Hdl = high-density lipoprotein; LDL = low-density lipoprotein.

## Data Availability

The datasets used during the current study are available from the corresponding author upon reasonable request.

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
