# Peer review of "Supervised Exercise in Water: Is It a Viable Alternative in Overweight/Obese People with or without Type 2 Diabetes? A Pilot Study"

_nutrients, 2022, doi:10.3390/nu14234963_

Round 1
Reviewer 1 Report (Previous Reviewer 2)
The authors have conveniently addressed all my comments and queries; thus, I recommend to accept the revised version 1 for publication at Nutrients.
Reviewer 2 Report (Previous Reviewer 1)
No further comment
This manuscript is a resubmission of an earlier submission. The following is a list of the peer review reports and author responses from that submission.
Round 1
Reviewer 1 Report
The authors present a pilot uncontrolled retrospective study (or a prospective study) which aims to give a contribute, showing the effects of a water-based exercise intervention, supervised by Graduates in Sports Sciences, in a group of overweight/obese people with or without type 2 diabetes.
Comments:
1.
The types of cohort studies(a observational study) are included:
Retrospective Cohort Study: The exposure and outcome information in a cohort study are identified retrospectively by using administrative datasets, reviewing patient charts, conducting interviews, etc. In a retrospective study, the outcome of interest has already occurred at the time the study is initiated.
Prospective Cohort Study: the investigator can establish a temporal relationship between an exposure and an outcome, or follow the natural development of a condition over time.
Longitudinal study, is also an observational study, in which data is gathered from the same sample repeatedly over an extended period of time.
This study design is?
2.
The study title “Exercise in water: is it a viable alternative in over-weight/obese people with or without type 2 diabetes”.
In this study, the over-weight/obese subjects are included type 2 diabetes patients and without type 2 diabetes.
The variations of anthropometric variables and clinical variables are higher in type 2 diabetes patients than in non-type 2 diabetes subjects.
How to calculate the relationships on the combined groups (with and without type 2 diabetes)?
3.
Characteristics of over-weight/obese with and without type 2 diabetes should be provided.
4.
How to calculate the repeated measure multivariate analysis of variance results?
Linear mixed models are a popular modelling approach for longitudinal or repeated measures data.
e.g.. Lancet 2014; 383: 1637–47, Statistical analysis section:
We used a mixed model repeated measures analysis to assess the mean profile over time for each treatment group. The fixed effects included in the linear model were: mean pain intensity at baseline, sex, age treatment, visit, and the interaction between treatment and visit. The random effect was patient………………….
5.
In this study, the timecourse of mean change in anthropometric variables and clinical variables from baseline, 3 months to 12 months of treatment(water-based exercise intervention) should be provided (Table 2).
6.
Difference of least square means (standard error, SE or 95% CI) should be provided on the change in anthropometric variables and clinical variables from baseline to 12 months (Table 2).
7.
The power test (Power and Sample Size Determination) should be calculated, due of a small sample sizes are recruited in this study.
Reviewer 2 Report
Review of Manuscript ID: nutrients-1931258
Title: Exercise in water: is it a viable alternative in over-weight/obese people with or without type 2 diabetes? A pilot uncontrolled retrospective study
Authors: Roberto Pippi, Matteo Vandoni, Matteo Tortorella, Vittorio Bini, Carmine Giuseppe Fanelli.
General comment: the study reports the beneficial effects of water exercise on Over-weight/obese subjects with or without type 2 diabetes mellitus. The authors submit a cohort of obese male/female humans to a moderate water exercise for 3 months, further for a year, and evaluate anthropometric parameters as well as blood pressure and lipid profile. The results show a significant decrease in body weight index, waist circumference, systolic and diastolic blood pressure, indicating that water-based exercise might be beneficial for the tested subjects. Despite the limitation of nutritional habits and pharmacological treatment, the hypothesis of increasing exercise with low risk of trauma is sound and attractive for subjects with or at risk of type 2 diabetes; methodology seems adequate and results show significant and promising changes. Some minor specific comments are detailed below:
Specific comments:
1) Line 111; it should be 25-meter.
2) Lines 163-164; there are several words in the text that are unnecessarily hyphenated, such as per-formed, statis-tics, that should be corrected through the whole text.
3) Lines 170-172 and tables 1 and 2; all dl should be dL, with capital L.
4) Line 206; acronym for type 2 diabetes mellitus should be uniform through the whole manuscript, DM2.
5) Lines 221-222; data of the mean decrease of SBP and DBP at three months are not exactly those reported in table 1
Round 2
Reviewer 1 Report
In Table 2, by authors statistical analysis: the estimated marginal means in our analysis of variance with a single factor -within subject- are the same as the observed real means.
However, the difference of distribution of a variable should be provided the estimated marginal means with standard error(SE) or 95% CI; it is not appropriate for the estimated marginal means with standard deviation(SD). This means there will be check the statistical signification.
The estimated delta changes should be provided the p values (T1 vs T0 or T2 vs T0; T1 vs T2 might be not important) and used method of multiple testing correction, e.g,. the Bonferroni adjustment.
A trend delta changes (from T0, T1, to T2) might be important, as β coefficient with SE or 95% CI. It is important analysis for intervention. How many to decline (trend of declining) by intervention of exercise in water?
The study design of “a pilot uncontrolled retrospective study” change to “a pilot longitudinal study” would be suggest.
